# Changes in Iron Status Biomarkers with Advancing Age According to Sex and Menopause: A Population-Based Study

**DOI:** 10.3390/jcm12165338

**Published:** 2023-08-16

**Authors:** Francesco Merlo, Dion Groothof, Farnaz Khatami, Noushin Sadat Ahanchi, Faina Wehrli, Stephan J. L. Bakker, Michele F. Eisenga, Taulant Muka

**Affiliations:** 1Institute of Social and Preventive Medicine, University of Bern, 3012 Bern, Switzerland; francesco.merlo@students.unibe.ch (F.M.); farnaz.khatami@unibe.ch (F.K.); noushin.sadat@unibe.ch (N.S.A.); 2Division of Nephrology, Department of Internal Medicine, University Medical Center Groningen, University of Groningen, Hanzeplein 1, 9713 GZ Groningen, The Netherlands; d.groothof@umcg.nl (D.G.); s.j.l.bakker@umcg.nl (S.J.L.B.); m.f.eisenga@umcg.nl (M.F.E.); 3Graduate School for Health Sciences, University of Bern, 3012 Bern, Switzerland; 4Community Medicine Department, Tehran University of Medical Sciences, Tehran 1417613151, Iran; 5Department of Internal Medicine, Internal Medicine, Lausanne University Hospital, Rue de Bugnon 21, 1005 Lausanne, Switzerland; 6Dr. Risch, Lagerstrasse 30, 9470 Buchs, Switzerland; faina.wehrli@gmail.com; 7Epistudia, Schanzenstrasse 4a, 3008 Bern, Switzerland

**Keywords:** iron biomarkers, age and menopausal status

## Abstract

Background: The risk of chronic diseases increases markedly with age and after menopause. An increase in bodily iron following menopause could contribute to this phenomenon of increased risk of chronic diseases. We aimed to investigate how various iron biomarkers change with advancing age, according to sex and menopausal status. Methods: We enrolled community-dwelling individuals with available information on ferritin, transferrin, iron, hepcidin, and soluble transferrin receptor levels from the Prevention of Renal and Vascular Endstage Disease study. The association of the iron biomarkers with age, sex, and menopausal status was investigated with linear regression models. Results: Mean (SD) age of the 5222 individuals (2680 women [51.3%], among whom 907 [33.8%] were premenopausal, 529 [19.7%] perimenopausal, and 785 [29.3%] postmenopausal), was 53.4 (12.0) years. Iron biomarkers showed a constant increase in women throughout their life course, in some cases at older ages surpassing values in men who, in turn, showed consistently higher levels of iron status compared to women in most age categories. Ferritin, hepcidin, and transferrin saturation levels were 3.03, 2.92, and 1.08-fold (all *p* < 0.001) higher in postmenopausal women compared to premenopausal. Conclusions: We found that iron accumulates differently depending on sex, age, and menopausal status. An increased iron status was identified in women, especially during and after menopause.

## 1. Introduction

Iron is the most abundant trace element in humans and is needed for oxygen and lipid metabolism, protein production, cellular respiration, and DNA synthesis. Iron gradually accumulates with age [1]. Evidence shows that bodily iron is generally higher in males than females and increases after menopause [2]. The risk of cardiovascular morbidity and mortality follows a similar pattern [3]. In 1981, Sullivan postulated the “iron hypothesis”, which states that the higher occurrence of cardiovascular morbidity and mortality in men and postmenopausal women versus premenopausal women could be due to higher iron stores in the former two groups, albeit findings have been inconclusive [4,5,6]. In addition, higher iron status is associated with an increased risk of type 2 diabetes and osteoporosis [7], which are age-dependent diseases and more common in postmenopausal women than premenopausal women. A recent large Mendelian randomization study has shown a causal harmful role of increased iron levels on mortality [8]. A further understanding of how levels of iron biomarkers differ with age, sex, and menopause status could therefore provide new insights into sex and menopausal disparities in cardiovascular disease. Only a few studies have investigated the premise that iron accumulates differently according to age, sex, and menopausal status, and those studies have been mainly focusing on ferritin levels. Boy infants had lower serum ferritin concentrations than girl infants [9]. However, in adulthood, men show higher levels of ferritin than women. Premenopausal women have lower ferritin levels than postmenopausal women [10]. Ferritin levels in women start to display a significant increase at the age of 40–49 years, and the levels continue to further increase from the age of 60 years, such that around the age of 90 years, women have achieved the same ferritin levels as men [1,10,11]. Hepcidin levels, a liver-derived peptide orchestrating systemic iron homeostasis, were shown to be lower in healthy women than in men [12]. While sex and menopausal differences in iron biomarkers are not fully understood, blood loss from menstruation has been implicated as the main causative factor. The high estrogen levels during the reproductive period in women could contribute to lower hepcidin mRNA expression and thus to lower levels of hepcidin [13]. Other factors, such as inflammation and lifestyle factors, could also contribute to these differences, which have not been taken into account by previous studies describing the association of sex and menopause with the levels of iron biomarkers [1,2]. Previous studies on iron accumulation after menopause have relied on age data rather than on menopausal state. This study aimed to quantify how various iron status parameters differentially accumulate with advancing age, according to sex and menopausal status in community-dwelling individuals enrolled in the Prevention of REnal and Vascular ENd-stage Disease (PREVEND) study.

## 2. Materials and Methods

### 2.1. Study Population and Design

The PREVEND study prospectively investigates risk factors for the prevalence and consequences of microalbuminuria in otherwise healthy adults in the city of Groningen (The Netherlands). The objectives and design have been described in detail elsewhere [14]. Briefly, all residents of Groningen, aged 28 to 75 years (n = 85,421), were invited to engage in the study between 1997 and 1998. With a response rate of 47.8%, pregnant women and patients with insulin-dependent diabetes were excluded. Participants with a urinary albumin concentration ≥ 10 mg/L (n = 6000) and a randomized control group with less than 10 mg/L (n = 2592) completed the first screening (n = 8592). The study consisted of five consecutive screening rounds, each screening comprising two visits to an outpatient clinic separated by three weeks. We finally included 5702 participants who attended the second screening and had available data on follow-up, hormone use, prior hysterectomy or ovariotomy, and iron status parameters. Detailed information on the flow of participants through the study is provided in Figure 1. To construct the table of clinical cohort characteristics, we reported the complete case samples (i.e., individuals with no missing observations on both outcome(s) and variable(s). Therefore, we excluded an additional 480 cases out of 5702 individuals (n = 5222).

The PREVEND study has been approved by the local medical ethics committee (MEC 96/01/022) and was undertaken in accordance with the Declaration of Helsinki. All participants provided written informed consent. Nine age categories for 5222 participants were created with 5-year intervals to compare iron levels by sex. For the analysis of iron levels by menopausal status, we excluded 440 women who had undergone hysterectomy, bilateral ovariotomy, or hormone replacement therapy due to menopause.

### 2.2. Menopausal Categories

To describe the change in iron levels by menopause status or reproductive stages, we divided the subjects into different categories based on their menopausal status. We defined menopause as a lack of menstruation for longer than 12 months. As the data on iron measures were obtained in the second screening round, the menopausal category refers to that time point. However, menopausal status was not assessed in the second screening round of PREVEND; therefore, we combined the menopause questionnaire available from the first and third surveys. In the first survey, we assessed whether women were currently menstruating and, if not, when was the last time. In the third survey, data were available on whether the women were currently menstruating. We defined not menstruating women in the first and third surveys as postmenopausal. With this information, we were able to create two certain and one uncertain menopausal categories for the second survey. Women being postmenopausal in both the first and the third surveys, were defined as postmenopausal in the second survey. Women who were premenopausal in both the first and the third surveys were classified as premenopausal in the second survey. We categorized women who were premenopausal in the first survey and postmenopausal in the third survey as “perimenopausal” women in menopausal transition since they could have been still premenopausal or already postmenopausal during the second survey. Perimenopause is defined as the transitional time with increasing episodes of amenorrhea before the final menstrual period and can last nearly 4 years for an average woman (McKinley et al.) [15]. In our study group, the second survey took place roughly 4 years after the first survey, and therefore, this uncertain perimenopausal status in the second survey, somewhere in between premenopause and postmenopause, is in line with the perimenopausal status defined by McKinley et al.

### 2.3. Iron Measurements

In the PREVEND study, five measures of iron status were measured (i.e., serum ferritin, transferrin, serum iron, hepcidin, and soluble transferrin receptor [sTfR]) from samples of the second survey. Fasting blood samples were obtained in the morning from all participants between 2001 and 2003. Aliquots of these samples were stored immediately at −80 °C until further analysis. An immunoassay and immunoturbidimetric assay were used to assess serum iron (µmol/L), ferritin (µg/L), and transferrin (g/L). The total iron binding capacity (TIBC) was calculated by multiplying transferrin by 25.2. Transferrin saturation [TSAT (%)] was derived by dividing serum iron by TIBC and multiplying by 100. Serum hepcidin concentrations were expressed in nanomoles per liter (nmol/L) and were measured with a competitive enzyme-linked immunosorbent assay (ELISA), as described elsewhere [16]. The between-plate and interassay coefficients of variation were 8.6% and 16.2%, respectively. sTfR was measured using an automated homogenous immunoturbidimetric assay with intra- and interassay CVs < 2% and <5% [17] 

### 2.4. Covariates

High-sensitivity C-reactive protein (hs-CRP) was also determined using nephelometry with a threshold of 0.175 mg/L and intra- and interassay coefficients of <4.4% and 5.7%, respectively. Smoking was defined as current smoking or smoking cessation within the previous year. Alcohol intake was assessed by self-reporting and classified as an intake of ≥10 g/day or lower. Minimum waist circumference was measured on bare skin at the natural indentation between the 10th rib and the iliac crest. When there was no indentation, we measured it in the middle between the navel and rib cage.

### 2.5. Statistical Analyses

To account for and reduce potential bias due to missing data, multiple imputations of incomplete covariates using fully conditional specification were performed to obtain 10 imputed datasets [18]. Analyses were performed in each dataset, and results were pooled using Rubin’s rules [19]. Baseline characteristics are shown for unimputed data (n = 5222) and are expressed as mean (standard deviation (SD)), median (interquartile range), or number (percentage) for normally distributed, skewed, and categorical data, respectively. Ferritin, hepcidin, and sTfR were log_2_-transformed prior to analysis to approximate normal distribution. Cross-sectional associations of age and sex with iron parameters (i.e., ferritin, TSAT, serum iron, hepcidin, and sTfR) were quantified based on imputed data (N = 5702) using linear regression models specifying the following variables as the main effects: specific iron parameter concerned, age, sex, current smoking, waist circumference, log_2_ hs-CRP, and alcohol consumption, which were selected based on previous literature or biological plausibility. Potential interactions between predictors were explored by introducing the following product terms into the model: age and sex, age and hs-CRP, and age and waist circumference. To allow for nonlinear effects of age, natural cubic splines with two degrees of freedom were used, wherein boundary knots were set to the 5th and 95th percentiles of age. Likelihood ratio tests were used to select an appropriate structure of the linear predictor, wherein *p* values were corrected for multiple testing according to the Bonferroni method. Model parameters, along with their confidence intervals and Wald test-based *p* values, were omitted from the main manuscript, as they do not have a straightforward interpretation (because of the interactions and nonlinear effects) but are presented in Appendix A. The results are visualized to facilitate their interpretation. The models’ assumptions were validated by plotting residuals against the fitted values and every predictor. The normality of the residuals was evaluated by inspection of Q-Q plots. Statistical analyses were performed with R version 4.0.2 (Vienna, Austria). A two-sided *p* < 0.05 was considered to indicate statistical significance.

## 3. Results

### 3.1. Baseline Characteristics

The baseline characteristics of the population included in the analyses (unimputed, n = 5222), stratified according to sex and menopause status, are shown in Table 1. The mean (SD) age was 54.3 (12.4) and 52.6 (11.5) years for men and women, respectively. Among 2680 (51.3%) female participants, 907 (35.6%) were premenopausal, 529 (20.7%) were perimenopausal, and 785 (30.8%) were postmenopausal. Men were older and reported higher levels of alcohol consumption, and had higher values of triglycerides, total cholesterol, waist circumference, systolic blood pressure, and diastolic blood pressure at baseline examination. Moreover, among women, the postmenopausal group were older and had higher values of triglycerides, total cholesterol, waist circumference, systolic blood pressure, and diastolic blood pressure. Also, postmenopausal women reported higher consumption of antidiabetic and lipid-lowering medications. Except for sTfR, all iron biomarkers, hemoglobin, and mean corpuscular volume were significantly higher in men than women (all *p* values < 0.001). Except for serum iron and sTfR, levels of ferritin, transferrin, and hepcidin were different across different menopausal groups of women.

In the overall study population, iron parameters were within the reference values, although the range in the population was wide (Table 1 and Table 2). The iron parameters and laboratory reference values of imputed data are shown in Table 2.

### 3.2. Iron Biomarkers by Sex and Age

The distribution of different iron measures by sex and age are shown in Appendix A. While levels of serum ferritin, TSAT, and hepcidin were higher in men of younger age compared to their women counterparts, the sex differences disappeared with advancing age. No difference in the level of sTfR was observed between men and women across different age categories. The multivariable analysis showed similar results. Compared with women, men showed higher levels of ferritin, TSAT, hepcidin, and serum iron but similar levels of sTfR (Appendix A). In further analyses, the associations of sex with ferritin, TSAT, hepcidin, and sTfR appeared to depend on age in such a way that the differences between men and women attenuate with advancing age (*p* interaction of age with sex < 0.05, Appendix A and Figure 2).

No significant interactions were observed between sex and age for serum iron (Appendix A). A nonlinear association of age with ferritin, hepcidin, and sTfR was observed (*p* for quadratic term < 0.05, Appendix A, Figure 2). Adjustment for confounders and exclusion of observations in iron biomarkers for which residuals were ≥2.58 SD below or above the mean residual did not substantially differ from the results from the main analysis (Appendix A). A graphic presentation of levels of iron biomarkers by age and sex using cubic splines is presented in Figure 2. Appendix A on the other hand, displays densities of various iron parameters according to age categories and sex.

### 3.3. Iron Biomarkers by Menopausal Status and Age

The distribution of different iron measures by menopause and age is visible in Figure 3. All measures indicate increased iron storage in postmenopausal women compared to other women categories. The statistical results are provided in Appendix A. Levels of ferritin, TSAT, and hepcidin increased with an advanced reproductive stage, with postmenopausal women having higher levels (Appendix A). However, the associations of menopause with ferritin and hepcidin depended on age (*p* value for interaction between menopause and age < 0.05, Appendix A), with age showing a nonlinear association with ferritin and hepcidin. In the premenopausal phase, independent of age, ferritin and hepcidin levels were constant (Figure 3).

In the perimenopause period, ferritin and hepcidin levels increased sharply with age, while this increase was milder in postmenopausal status (Figure 3). The level of TSAT enhanced significantly from pre- to peri- and from peri- to postmenopausal status, and in line with this finding, sTfR level significantly declined from pre- to peri- and from peri- to postmenopausal status (Appendix A). No significant differences were observed for serum iron either from pre- to perimenopause or from peri- to postmenopause. Exclusion of observations in iron biomarkers for which residuals were ≥2.58 SD below or above the mean residual did not change the results (Appendix A).

## 4. Discussion

In this population-based cohort study, we found that the levels of iron biomarkers differ over the life course, depending on sex and menopausal status. The levels of ferritin and hepcidin reflect higher iron stores in males than in females at a younger age, but after the age of 55, these differences even out. Also, levels of ferritin, TSAT, and hepcidin show that women have higher iron stores after their menopausal transition than before. No difference by sex was found for TSAT, sTfR, and no difference by menopausal status for sTfR. The various iron biomarkers indicate an increasing iron level in women throughout their lifespan, dynamics which are absent in men, who, in general, have higher iron levels than women. The increase in iron levels in women occurs mainly during perimenopause, while in postmenopausal women, the increase is less pronounced. Our results on ferritin being higher in men than in women and its changes by age are consistent with others [1,20,21]. Previous cross-sectional studies have shown that women of reproductive age have lower ferritin levels than middle-aged and older women [22,23]. In one study, serum ferritin levels were examined by age, sex, and race in 20,040 individuals > 17 years of age, and it was shown that ferritin levels remained relatively low until menopause, after which they rose but not to levels as high as in men [1]. Our study extends previous findings showing not only differences in ferritin levels but also in other iron status parameters between men and women and by menopausal status, independent of potential confounding factors [2]. In addition, previous studies used age to define menopause. In our study, hepcidin, the key regulator of iron metabolism, was lower in women than in men, but this difference was attenuated with age, such that hepcidin levels were higher in older women than in men of the same age category. Little evidence exists on levels of hepcidin during the life course [24] and how these levels differ by sex. Jaeggi et al. found in 339 Kenyan infants aged 6.0 ± 1.1 months that serum hepcidin levels were significantly lower in male infants than in females [25]. However, Ganz et al. reported in 114 healthy adults that serum hepcidin levels were lower in women than in men [12]. These data could indicate that differences in levels of hepcidin and other iron biomarkers may start early in life and that sex-specific factors such as puberty/menarche or menopause could have an impact. A comprehensive investigation of levels of iron biomarkers across the life course would provide more insight. TSAT is a measure of the number of iron molecules that are bound to transferrin [26]. We found that TSAT behaves similarly to ferritin and hepcidin in terms of age and sex. Cook et al., in a study of 1564 subjects living in the northwestern United States, showed that age and sex in determining TSAT levels were less pronounced than for ferritin. The median values of TSAT varied relatively little, from 22% in children to 28% in young adult males [21]. In addition, our results have shown that TSAT enhances significantly from pre- to peri- and from peri- to postmenopausal status. Levels of sTfR were the same in both sexes. sTfR, a promising marker of unmet cellular iron demands and erythroid activity, is proved as an indicator of early iron deficiency in the general population [27]. In our study, sTfR levels significantly declined from pre- to peri- and from peri- to postmenopausal status, in line with a previous longitudinal study showing decreased sTfR levels from pre- to post-menopause [2]. Menstrual blood loss, pregnancies, and deliveries could explain why young women fail to acquire significant iron stores [20]. Cessation of menstruation marks the later stage of perimenopause, and because iron is no longer lost through menstruation, it may accumulate in the body. Similar increased iron levels are also observed during pregnancy or in women with polycystic ovary syndrome, who have lack or irregular menstrual periods. Moreover, serum ferritin and hepcidin are acute-phase proteins, and their levels are altered with increased inflammation [21]. Thus, whether the increased ferritin and hepcidin levels with menopause and age in women reflect an actual increase in iron stores or an upregulation due to inflammation should be further investigated. However, in our analysis, we adjusted for hs-CRP levels and removed participants with chronic inflammation, suggesting that there is an increase in iron stores.

### Strengths and Limitations

The strengths of our study were the number of different biomarkers and the large population size of healthy participants. While other studies often focused only on one iron status measure and did not focus on menopausal status, we could explore iron status by age and sex using five different iron biomarkers. Limitations of the study include having participants restricted to one city, so it remains unclear if the results are generalizable to other population groups. The cyclic variation in iron is a possible source of error when iron status is assessed in a large population, but for nonmenopausal women, we did not have information on which day of the menstrual cycle iron was measured, and we could not take this into account in our analysis. Another limitation is the definition of menopause, which is self-reported, and it was not assessed in the visit where iron biomarkers were measured. Thus, misclassification of menopause status may be present. In addition, iron status can be affected by diet. In our study, we did not have information on nutritional intake, and thus residual confounding could be present. Nevertheless, adjustment for alcohol intake did not affect our findings. Furthermore, in a previous publication, we have shown that menopause status is not associated with changes in diet [28]. Thus, our findings on menopause status and iron biomarkers are most likely not to be confounded by diet. Another limitation is that we did not have information available about blood donation history, which can have a significant effect on iron status and could have affected the results of our study. Finally, the observational design and cross-sectional nature of the study might have biased the results due to the residual confounding, and our observations do not reflect causal associations.

## 5. Conclusions

In conclusion, this study showed that iron accumulates differently among men and women within different age categories and in women within different menopausal categories. These findings suggest that sex and menopause-specific cut-offs of iron biomarkers should be explored and determined and the current sex-specific cut-offs may be biased. In addition, understanding the health impact of increased iron body stores, especially during perimenopause in women, could be important for disease prevention. Iron is associated with several health outcomes, including cardiovascular disease, diabetes, depression, and osteoporosis, all of which are more prominent with age and during postmenopausal years in women. Phlebotomy, a procedure aiming at reducing iron body stores, has been shown to be as effective as oral contraceptives in improving insulin resistance in polycystic ovary syndrome, as well as decreasing insulin resistance in diabetes [29,30,31]. Future longitudinal studies with repeated measures of iron biomarkers could better explore the trajectories of iron by sex, age, and menopause and how these trajectories relate to different health domains.

## Figures and Tables

**Figure 1 jcm-12-05338-f001:**
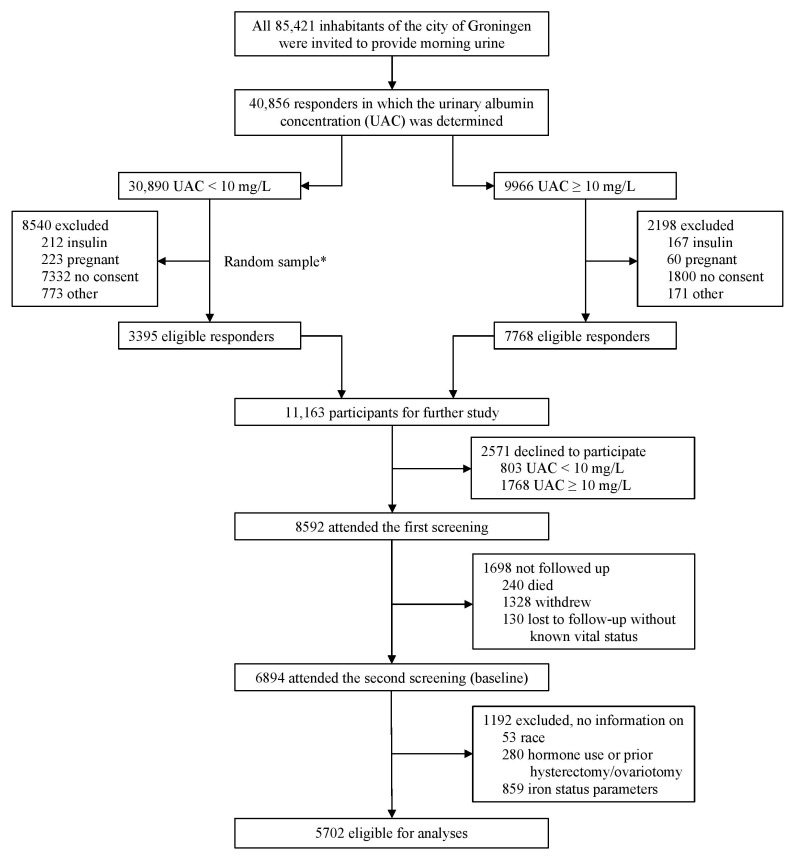
Flow of the participants through the study. * Size of the random sample was arbitrarily set at 3395 (out of the 22,350 eligible participants) to obtain a total cohort size of approximately 10,000, considering a 15% nonparticipation rate.

**Figure 2 jcm-12-05338-f002:**
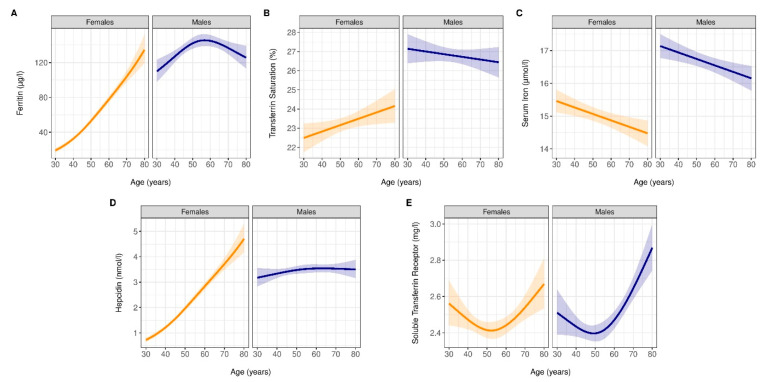
Concentrations of iron parameters with advancing age and across sex. (**A**), Effects of age and sex on ferritin. (**B**), Effects of age and sex on transferrin saturation. (**C**), Effects of age and sex on serum iron. (**D**), Effects of age and sex on hepcidin. (**E**), Effects of age and sex on soluble transferrin receptor (sTfR). The lines represent the expected concentrations: orange for females and blue for males. These were derived from linear regression models. For ferritin, transferrin saturation, hepcidin, and sTfR, due to statistically significant interaction between age and sex, the shapes of the lines are different between females and males. For serum iron, where no statistically significant interaction was observed, the shapes of the lines are identical between females and males. The shaded areas about the lines indicate the corresponding 95% pointwise confidence intervals. Ferritin, hepcidin, and sTfR were log_2_ transformed.

**Figure 3 jcm-12-05338-f003:**
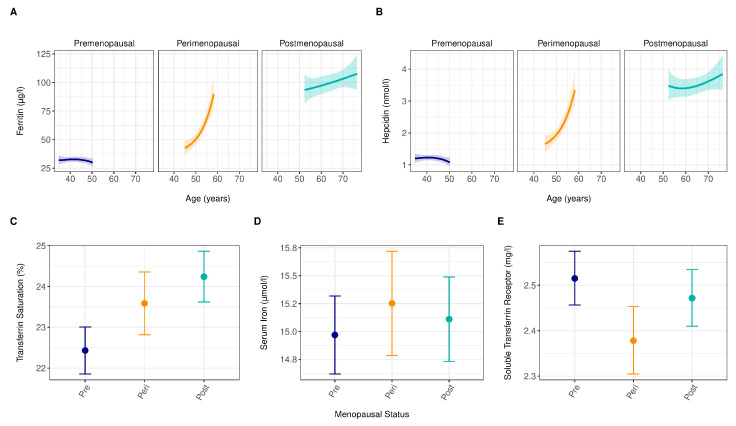
Concentrations of iron parameters with advancing age and across menopausal status. (**A**), Associations between ferritin levels with age and menopausal status. (**B**), Associations between hepcidin with age and menopausal status. (**C**), Associations between transferrin saturation with age and menopausal status. (**D**), Associations between serum iron with age and menopausal status. (**E**), Associations between soluble transferrin receptor (sTfR) with age and menopausal status. The lines (**A**,**B**) and dots (**C**–**E**) represent expected concentrations: blue for premenopausal, orange for perimenopausal, and green for postmenopausal. These were derived from linear regression models. For ferritin and hepcidin, due to a statistically significant interaction between menopausal status and age, expected concentrations are shown for the central 90% age range within each menopausal category, with shaded areas indicating the corresponding 95% pointwise confidence intervals. For transferrin saturation, serum iron, and sTfR, where no statistically significant interaction was observed, expected concentrations are shown as dots at the median age within each menopausal status, with whiskers indicating the corresponding 95% confidence intervals. Ferritin, hepcidin, and sTfR were log_2_ transformed.

**Table 1 jcm-12-05338-t001:** Characteristics of the Unimputed Cohort Data at Baseline.

					Menopausal Status ^†^			
	Total Population (n = 5222)	Men (n = 2542)	Women (n = 2680)	*p* for Difference	Premenopausal (n = 907)	Perimenopausal (n = 529)	Postmenopausal (n = 785)	*p* for Trend
Sociodemographic characteristics								
Age, mean (SD), years	53.4 (12.0)	54.3 (12.4)	52.6 (11.5)	<0.001	42.1 (4.8)	52.1 (4.2)	64.0 (7.2)	<0.001
Categories of age, no. (%) *, years								
<40	768 (14.7)	369 (14.5)	399 (14.9)	<0.001	326 (35.9)	5 (0.9)	0 (0.0)	<0.001
40–50	1502 (28.8)	679 (26.7)	823 (30.7)		531 (58.5)	146 (27.6)	15 (1.9)	
50–60	1439 (27.6)	664 (26.1)	775 (28.9)		50 (5.5)	366 (69.2)	246 (31.3)	
60–70	897 (17.2)	464 (18.3)	433 (16.2)		0 (0.0)	12 (2.3)	354 (45.1)	
>70	616 (11.8)	366 (14.4)	250 (9.3)		0 (0.0)	0 (0.0)	170 (21.7)	
Race, no. (%)								
Caucasian	5016 (96.3)	2434 (95.8)	2582 (96.3)	0.43	859 (94.7)	513 (97.0)	780 (99.4)	<0.001
Negroid	46 (0.9)	21 (0.8)	25 (0.9)		12 (1.3)	4 (0.8)	1 (0.1)	
Asian	103 (2.0)	54 (2.1)	49 (1.8)		23 (2.5)	9 (1.7)	4 (0.5)	
Other	57 (1.1)	33 (1.3)	24 (0.9)		13 (1.4)	3 (0.6)	0 (0.0)	
Education, no. (%)								
Low	2248 (43.0)	983 (38.7)	1265 (47.2)	<0.001	236 (26.0)	238 (45.0)	548 (69.8)	<0.001
Middle	1324 (25.4)	704 (27.7)	620 (23.1)		270 (29.7)	119 (22.5)	120 (15.3)	
High	1650 (31.6)	855 (33.6)	795 (29.7)		401 (44.2)	172 (32.5)	117 (14.9)	
Current smoking, no. (%)	1446 (27.7)	692 (27.2)	754 (28.1)	0.46	256 (28.2)	151 (28.5)	191 (24.3)	0.08
Alcohol consumption, ≥10 g/day, no. (%)	1381 (26.4)	847 (33.3)	534 (19.9)	<0.001	178 (19.6)	128 (24.2)	159 (20.3)	0.70
Prevalent cardiovascular disease, no. (%)	313 (6.0)	223 (8.8)	90 (3.4)	<0.001	15 (1.7)	14 (2.6)	38 (4.8)	<0.001
Prevalent type 2 diabetes, no. (%)	291 (5.6)	167 (6.6)	124 (4.6)	0.002	7 (0.8)	18 (3.4)	66 (8.4)	<0.001
Waist circumference, mean (SD), cm	91.8 (12.7)	96.9 (11.0)	87.0 (12.2)	<0.001	83.0 (11.1)	87.1 (11.7)	91.2 (12.2)	<0.001
Hemodynamics								
Systolic blood pressure, mean (SD), mm Hg	125.7 (18.6)	130.0 (17.6)	121.5 (18.6)	<0.001	113.6 (12.7)	120.4 (17.3)	130.1 (20.3)	<0.001
Diastolic blood pressure, mean (SD), mm Hg	73.1 (9.0)	76.0 (8.6)	70.4 (8.5)	<0.001	67.9 (8.0)	71.7 (8.7)	72.4 (8.3)	<0.001
Lipid spectrum								
Total cholesterol, mean (SD), mmol/L	5.4 (1.0)	5.4 (1.0)	5.5 (1.1)	0.23	4.9 (0.9)	5.7 (1.0)	5.9 (1.0)	<0.001
HDL cholesterol, mean (SD), mmol/L	1.3 (0.3)	1.1 (0.3)	1.4 (0.3)	<0.001	1.4 (0.3)	1.4 (0.3)	1.4 (0.3)	0.28
Total cholesterol/HDL cholesterol, mean (SD)	4.5 (1.3)	5.0 (1.3)	4.1 (1.1)	<0.001	3.7 (1.0)	4.2 (1.1)	4.4 (1.1)	<0.001
Triglycerides, median (IQR), mmol/L	1.1 (0.8 to 1.6)	1.2 (0.9 to 1.8)	1.0 (0.7 to 1.4)	<0.001	0.8 (0.6 to 1.1)	1.1 (0.8 to 1.4)	1.2 (0.9 to 1.6)	<0.001
Haematologic parameters								
Haemoglobin, mean (SD), mmol/L	8.5 (0.8)	9.0 (0.6)	8.1 (0.6)	<0.001	7.9 (0.6)	8.1 (0.6)	8.2 (0.6)	<0.001
Mean corpuscular volume, mean (SD), fl	90.5 (4.6)	90.9 (4.3)	90.1 (4.8)	<0.001	89.7 (5.2)	90.6 (4.2)	90.4 (4.6)	0.001
Inflammation								
High-sensitivity C-reactive protein, median (IQR), mg/L	1.3 (0.6 to 2.9)	1.3 (0.6 to 2.7)	1.4 (0.6 to 3.2)	>0.99	1.0 (0.4 to 2.8)	1.3 (0.7 to 2.7)	1.8 (0.9 to 3.6)	<0.001
Renal function parameters								
Cystatin C-based eGFR, mean (SD), mL/min per 1.73 m^2^	90.4 (19.4)	89.7 (20.2)	91.0 (18.5)	0.013	102.5 (13.1)	92.7 (14.6)	79.1 (16.6)	<0.001
Urinary albumin excretion, mg/day	8.3 (6.0 to 14.4)	9.3 (6.5 to 17.8)	7.5 (5.6 to 11.8)	<0.001	7.1 (5.5 to 10.7)	7.8 (5.7 to 12.1)	8.1 (5.7 to 13.3)	<0.001
Categories of urinary albumin excretion, no. (%)								
<15 mg/day	3981 (76.1)	1775 (69.8)	2206 (82.3)	<0.001	787 (86.8)	433 (81.9)	613 (78.1)	<0.001
15–29.9 mg/day	642 (12.3)	359 (14.1)	283 (10.6)		76 (8.4)	58 (11.0)	98 (12.5)	
30–300 mg/day	531 (10.2)	355 (14.0)	176 (6.6)		37 (4.1)	37 (7.0)	71 (9.0)	
>300 mg/day	68 (1.3)	53 (2.1)	15 (0.6)		7 (0.8)	1 (0.2)	3 (0.4)	
Iron Parameters								
Ferritin, median (IQR), µg/L	97 (48 to 172)	145 (87 to 232)	60 (30 to 113)	<0.001	33 (17 to 61)	61 (35 to 106)	106 (63 to 161)	<0.001
Transferrin saturation, mean (SD), %	25.0 (9.4)	26.8 (9.2)	23.4 (9.2)	<0.001	22.6 (10.7)	23.7 (8.2)	24.4 (8.0)	<0.001
Serum iron, mean (SD), µmol/L	15.8 (5.6)	16.6 (5.5)	15.1 (5.6)	<0.001	15.0 (6.7)	15.3 (4.9)	15.2 (4.6)	0.55
Hepcidin, median (IQR), nmol/L	3.0 (1.7 to 4.9)	3.8 (2.4 to 5.6)	2.4 (1.2 to 4.1)	<0.001	1.3 (0.6 to 2.3)	2.6 (1.5 to 3.9)	3.7 (2.4 to 5.6)	<0.001
Soluble transferrin receptor, median (IQR), mg/L	2.47 (2.08 to 2.97)	2.45 (2.05 to 2.96)	2.50 (2.11 to 2.97)	0.001	2.48 (2.01 to 3.08)	2.37 (2.00 to 2.84)	2.48 (2.12 to 2.94)	0.52
Medication								
Antihypertensive drugs, no. (%)	1083 (20.7)	578 (22.7)	505 (18.8)	0.001	59 (6.5)	93 (17.6)	256 (32.6)	<0.001
Lipid-lowering drugs, no. (%)	485 (9.3)	285 (11.2)	200 (7.5)	<0.001	22 (2.4)	21 (4.0)	118 (15.0)	<0.001
Hormones for climacteric, no. (%)	-	-	97 (4)		3 (0)	51 (10)	29 (4)	<0.001
Hormones for other reasons, no. (%)	-	-	68 (3)		20 (2)	14 (3)	21 (3)	0.49

* Percentages may not total 100 because of rounding. Abbreviations: eGFR, estimated glomerular filtration rate; SD, standard deviation; IQR, interquartile range. ^†^ Listwise deletion was used to remove cases with missing observations in the menopausal status.

**Table 2 jcm-12-05338-t002:** Iron Parameters of the imputed Cohort Data and Laboratory Reference Values.

			Menopausal Status ^†^			
Iron Parameters *	Total Population	Premenopausal (n = 989)	Perimenopausal (n = 555)	Postmenopausal (n = 849)	*p* for Trend	Reference Values
Ferritin, median (IQR), µg/L	97 (47 to 171)	33 (17 to 61)	61 (35 to 105)	106 (63 to 162)	<0.001	♂ 30–400; ♀ 15–130
Transferrin saturation, mean (SD), %	25.0 (9.4)	22.4 (10.7)	23.6 (8.2)	24.2 (8.0)	<0.001	♂ 16–45%; ♀ 14–35
Serum iron, mean (SD), µmol/L	15.8 (5.6)	15.0 (6.7)	15.3 (4.9)	15.1 (4.6)	0.61	♂ 14–35; ♀ 10–25
Hepcidin, median (IQR), nmol/L	3.0 (1.7 to 4.9)	1.3 (0.6 to 2.3)	2.6 (1.5 to 3.8)	3.8 (2.4 to 5.7)	<0.001	♂ 0.5–14.7; ♀ 0.5–14.6
Soluble transferrin receptor, median (IQR), mg/L	2.47 (2.09 to 2.97)	2.47 (2.01 to 3.08)	2.38 (2.02 to 2.84)	2.47 (2.11 to 2.95)	0.61	♂ 2.2–5; ♀ 1.9–4.4

* Laboratory reference values from the study institute (UMCG, University Medical Center Groningen). A log_2_ transformation was applied to circulating concentrations of ferritin, hepcidin, and sTfR to obtain approximate Gaussian distributions. ^†^ Listwise deletion was used to remove cases with missing observations in the menopausal status.

## Data Availability

The data of this study are available from the corresponding author to qualified investigators upon reasonable request.

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
