# Peer review of "Changes in Iron Status Biomarkers with Advancing Age According to Sex and Menopause: A Population-Based Study"

_jcm, 2023, doi:10.3390/jcm12165338_

Round 1
Reviewer 1 Report
Summary- The authors of this study have performed a population-based analysis of iron storage biomarkers within a subgroup of individuals participating in a larger study conducted in a specific city. Alongside recording lifestyle markers and variables like inflammation indicator, which could potentially affect blood iron status, the authors focused on exploring the correlation between iron biomarkers and key demographic factors, including age, sex, and menopausal status.
General comments- The overall impact of the study on the current knowledge and already known factors impacting the iron status is very limited.
Specific comments-
- In the introduction, it would be valuable to include references to other relevant studies that have investigated the relationship between menopausal status and iron biomarkers. This would provide a broader context and demonstrate the existing knowledge in the field regarding this particular correlation.
Example: Anna. Hematol. N. Milman et. al. 1992.
- Throughout the paper, while discussing iron overload and its impact on diseases, it would be advantageous to show how your data corelates to references supporting the potential role of iron status as a marker for the onset of diseases like type 2 diabetes/ or if you did not find any such corelation. This would strengthen the study by considering additional correlations and providing a more comprehensive understanding of the implications of iron biomarkers.
- In the section discussing the limitations and strengths of the study (4.1), it is important to address certain factors that may impact the interpretation of the results. Specifically, information on the diet of the subjects, especially if any of them were vegetarian, should be included if available. Additionally, it should be explicitly stated whether any of the subjects were blood donors and if that information is not available.
Furthermore, it is worth acknowledging the potential influence of alcohol intake on blood iron levels. Studies have shown that even small amounts of alcohol consumption can impact iron metabolism. Given that a significant proportion of the subjects in this study use alcohol, it is essential to consider the potential confounding effect of alcohol on your findings.
Author Response
Response to the reviewers:
Reviewer 1:
Point 1: In the introduction, it would be valuable to include references to other relevant studies that have investigated the relationship between menopausal status and iron biomarkers. This would provide a broader context and demonstrate the existing knowledge in the field regarding this particular correlation.
Example: Anna. Hematol. N. Milman et. al. 1992.
Response 1: We thank the reviewer for the suggestion. We now cited more studies that investigated the link between menopause and iron status, including the reference suggested by the reviewer.
Page 2, line 56- 59 “Premenopausal women have lower ferritin levels than postmenopausal women [1]. Ferritin levels in women start to display a significant increase at the age of 40–49 years and the levels continue to further increase from the age of 60 years, such that around the age of 90 years old, women have achieved the same ferritin levels as men [1-3].”
Point 2: Throughout the paper, while discussing iron overload and its impact on diseases, it would be advantageous to show how your data correlates to references supporting the potential role of iron status as a marker for the onset of diseases like type 2 diabetes/ or if you did not find any such correlation. This would strengthen the study by considering additional correlations and providing a more comprehensive understanding of the implications of iron biomarkers.
Response 2: We thank the reviewer for the comment. In this study, we did not investigate the association between diabetes and iron biomarkers, of which we agree that it is an important topic to explore and deserving more attention than could be paid in a manuscript that focuses on iron status and menopausal status. We are currently working on two manuscripts exploring the impact of iron on diabetes, which we plan to publish after this first paper.
Point 3: In the section discussing the limitations and strengths of the study (4.1), it is important to address certain factors that may impact the interpretation of the results. Specifically, information on the diet of the subjects, especially if any of them were vegetarian, should be included if available. Additionally, it should be explicitly stated whether any of the subjects were blood donors and if that information is not available.
Furthermore, it is worth acknowledging the potential influence of alcohol intake on blood iron levels. Studies have shown that even small amounts of alcohol consumption can impact iron metabolism. Given that a significant proportion of the subjects in this study use alcohol, it is essential to consider the potential confounding effect of alcohol on your findings.
Response 3: We share the reviewer’s point of view. In this study, we did not have the nutritional status and history of blood donation of the participants available. As such, we could unfortunately not explore how these factors might have impacted our findings. We have added mentioning of the lack of these variables to the limitations section of the revised version of our manuscript. Nevertheless, for the revised version of the manuscript, we now take into account the role of alcohol intake, which after adjustment did not materially alter our results. In addition, in a recent publication we have shown that diet, including alcohol consumption, does not change by menopause status. Therefore, we consider our results by menopause status robust despite lack of adjustment for nutritional data.
Page 10, line 315- 323 “In addition, iron status can be affected by diet. In our study, we did not have information on nutritional intake, and thus residual confounding could be present. Nevertheless, adjustment for alcohol intake did not affect our findings. Furthermore, in a previous publication, we have shown that menopause status is not associated with changes in diet [4]. Thus, our findings on menopause status and iron biomarkers are most likely not to be confounded by diet. Another limitation is that we did not have information available about blood donation history, which can have a significant effect on iron status and could have affected the results of our study.”
- Milman, N., M. Kirchhoff, and T. Jørgensen, Iron status markers, serum ferritin and hemoglobin in 1359 Danish women in relation to menstruation, hormonal contraception, parity, and postmenopausal hormone treatment. Ann Hematol, 1992. 65(2): p. 96-102.
- Zacharski, L.R., et al., Association of age, sex, and race with body iron stores in adults: analysis of NHANES III data. Am Heart J, 2000. 140(1): p. 98-104.
- Milman, N. and M. Kirchhoff, Iron stores in 1359, 30- to 60-year-old Danish women: evaluation by serum ferritin and hemoglobin. Ann Hematol, 1992. 64(1): p. 22-7.
- Grisotto, G., et al., Menopausal Transition Is Not Associated with Dietary Change in Swiss Women. J Nutr, 2021. 151(5): p. 1269-1276.

Reviewer 2 Report
The manuscript described study of iron metabolism in men and women with special focus on pre- and post-menopausal women. The authors have done a difficult job, but the presentation of the data and their discussion raise questions - and there are doubts about the adequacy of the results. A close reading of the materials and methods, and results sections raised the following questions:
1. The objectives of the study and study design should be briefly explained in the manuscript itself.
2. There were 5702 subjects available for analysis as per figure 1 but most of results were based on 5222. It was not explained how and why the subjects were omitted from study.
“From 8,592 individuals in first survey, 1,698 did not participate in second survey and 350 participants taking oral iron supplements were excluded. Nine age categories for 5,222 participants were created with 5-year intervals to compare iron levels by sex.”
8592-1698-350=6544. How does it become 5222.
In line 149, the number used is 5825 for linear regression models.
These numbers keep varying throughout the manuscript and need an explanation from authors.
3. Similarly, “2,680 women [51.3%] amongst whom 907 [33.8%] were premenopausal, 529 [19.7%] perimenopausal, and 785 [29.3%] postmenopausal”.
907+529+785= 2221. What happened to rest of subjects?
And it is like that for all groups as well in Table 1, there is a discrepancy throughout.
4. Did the authors use median for urinary albumin excretion? What was the range?
The range used is 6.0-14.4 mg/day and then subcategories have range >300mg/day. If they have used IQR, then it seems highly data is highly biased to one side.
5. The median and IQR values keep changing between Table 1 and Table 2.
For example: Ferritin in Table 1: 97 (48-172); ferritin in Table 2: 97 (48-171).
Hepcidin for post menopausal in Table 1: 3.7 (2.4-5.6); in Table 2: 3.8 (2.4-5.7) and so on.
6. In figure 3A and 3B, the word “transition” is used while in figure 3C-E and throughout manuscript, the word “perimenopausal” is used. It would be great to keep the homogeneity throughout manuscript to avoid confusion.
7. Figure 3 legend lacks clarity. Also, did the users did a statistical analysis between groups? If yes, a p-value or something like asterisk mark should be shown in figure.
8. Line 236, “No differences in serum iron were observed by menopausal status.” It is hard to grasp this statement in absence of a statistical analysis. Please explain with statistical analysis.
9. Line 253, “Previous cross-sectional studies have shown that women of reproductive age have lower iron biomarkers than middle-aged and elderly women [19, 20].” This statement is not clear because not all bio markers were low in women of reproductive age. For example: soluble transferrin receptor (figure 3E).
10. The authors have rightly mentioned in limitations in line 302, “we did not have information on which day of the menstrual cycle iron was measured and we could not take this into account in our analysis.” As iron levels can change drastically in menstruating women, this is an important parameter whether the blood was drawn before or after mensuration and the day of blood collection should have been ideally kept constant throughout study.
Additionally, I would like to understand why authors didn’t use the new age technology like ICPMS or AAS for serum iron measurement.
The English needs to be improved to improve the overall quality of manuscript and to make it readable. It is really difficult to interpret why authors want to say at some points.
Author Response
Response to the reviewers:
Reviewer 2:
Point 1: The objectives of the study and study design should be briefly explained in the manuscript itself.
Response 1: We thank the reviewer for this comment. Accordingly, we added a summary of information to the study population and design section.
Page 2, line 78- 82 “Briefly, all residents of Groningen, aged 28 to 75 years (n=85,421), were invited to engage in the study between 1997 and 1998. With a response rate of 47.8%, pregnant women and patients with insulin-dependent diabetes were excluded. Participants with a urinary albumin concentration ≥10 mg/L (n=6,000) and a randomized control group with less than 10 mg/L (n=2,592), completed the first screening (total subjects=8592).”
Point 2: There were 5702 subjects available for analysis as per figure 1 but most of results were based on 5222. It was not explained how and why the subjects were omitted from study.
“From 8,592 individuals in first survey, 1,698 did not participate in second survey and 350 participants taking oral iron supplements were excluded. Nine age categories for 5,222 participants were created with 5-year intervals to compare iron levels by sex.”
8592-1698-350=6544. How does it become 5222.
In line 149, the number used is 5825 for linear regression models.
These numbers keep varying throughout the manuscript and need an explanation from authors.
Response 2: We thank the reviewer for highlighting the typos. We have now revised the numbers and provided information for each step. In brief, a total of 8,592 individuals were enrolled to participate in the first screening round. Indeed, 1698 did not participate in the second survey and were excluded (n=6,894). Of these, 1,192 participants did not have information on race, hormone use or prior hysterectomy/ovariotomy, and iron status parameters, and finally, 5,702 participants were included for multiple imputation. To construct the table of clinical cohort characteristics, we reported the complete case samples (i.e., individuals with no missing observations on both outcome(s) and variable(s). Therefore, we excluded additional 480 cases out of 5,702 individuals (n=5,222). We revised the text to clear the ambiguity. In addition, we correct the number 5,828 for linear regression to 5,702.
Page 2, line 84- 89 “We finally included 5,702 participants who attended the second screening and had available data on follow up, hormone use, prior hysterectomy or ovariotomy, and iron status parameters. Detailed information on the flow of participants through the study is provided in Figure 1. To construct the table of clinical cohort characteristics, we reported the complete case samples (i.e., individuals with no missing observations on both outcome(s) and variable(s) (n=5,222).”
Page 4, line 155- 158 “Cross-sectional associations of age and sex with iron parameters (i.e., ferritin, TSAT, serum iron, hepcidin, and sTfR) were quantified based on imputed data (N=5,702), using linear regression models specifying the following variables as main effects”
Point 3: Similarly, “2,680 women [51.3%] amongst whom 907 [33.8%] were premenopausal, 529 [19.7%] perimenopausal, and 785 [29.3%] postmenopausal”.
907+529+785= 2221. What happened to rest of subjects?
And it is like that for all groups as well in Table 1, there is a discrepancy throughout.
Response 3: We thank the reviewer for this excellent comment. The reason for the discrepancy in the numbers is due to missing observations in menopausal status. In the title of Table 1, we mentioned that the baseline characteristics refer to unimputed cohort data (there are still missing observations in covariates). Therefore, the sum of females in each menopausal category does not equal the total number of females. We added more explanation about it in footer of the table 1.
Page 6, line 193 and page 7, line 202 “†Listwise deletion was used to remove cases with missing observations in the menopausal status.”
Point 4: Did the authors use median for urinary albumin excretion? What was the range?
The range used is 6.0-14.4 mg/day and then subcategories have range >300mg/day. If they have used IQR, then it seems highly data is highly biased to one side.
Response 4: We thank the rewiever for the comment. Yes, we used the median and IQR. Summary statistics of 24-hour urinary albumin excretion: Min: 2.17 mg/day; 25th percentile: 5.95 mg/day; median: 8.33 mg/day; mean: 28.98 mg/day; SD: 140.58 mg/day; 75th percentile: 14.37 mg/day; max: 4240.80 mg/day.
Point 5: The median and IQR values keep changing between Table 1 and Table 2. For example: Ferritin in Table 1: 97 (48-172); ferritin in Table 2: 97 (48-171).
Hepcidin for post-menopausal in Table 1: 3.7 (2.4-5.6); in Table 2: 3.8 (2.4-5.7) and so on.
Response 5: We thank the reviewer for this consideration. Table 2, as mentioned in the text, is related to the descriptive analysis after imputation, and therefore it is slightly different from the values in Table 1 (unimputed). We added the “imputation” statement for the title of Table 2.
Page 6, line 196- 197 “The iron parameters and laboratory reference values of imputed data are shown in Table 2.”
Page 7, line 198 “Table 2. Iron Parameters of the imputed Cohort Data and Laboratory Reference Values.”
Point 6: In figure 3A and 3B, the word “transition” is used while in figure 3C-E and throughout manuscript, the word “perimenopausal” is used. It would be great to keep the homogeneity throughout manuscript to avoid confusion.
Response 6: We thank the reviewer comment. Accordingly, we revised all “transitions” to “perimenopause” as defined in method section 2.2 for menopausal Categories.
Page 8, line 243; page 8, line 260; page 10, line 332 and Table S2 as a new supplementary file.
Point 7: Figure 3 legend lacks clarity. Also, did the users did a statistical analysis between groups? If yes, a p-value or something like asterisk mark should be shown in figure.
Response 7: We thank the reviewer for the comment. We revised the legend of Figure 3. In addition, the statistical results are provided in Supplementary Table S2.
Page 8, line 237- 242 “Figure 3. Concentrations of iron parameters with advancing age and across menopausal status. A, Associations between ferritin levels with age and menopausal status. B, Associations between hepcidin with age and menopausal status. C, Associations between transferrin saturation with age and menopausal status. D, Associations between serum iron with age and menopausal status. E, Associations between sTfR with age and menopausal status.
Ferritin, hepcidin, and sTfR were binary log-2 transformed.
Page 8, line 229 “The statistical results are provided in Supplementary Table S2.”
Point 8: Line 236, “No differences in serum iron were observed by menopausal status.” It is hard to grasp this statement in absence of a statistical analysis. Please explain with statistical analysis.
Response 8: We thank the reviewer for the comment. We added the statistical information.
Page 8, line 247- 248 “No significant differences were observed for serum iron either from pre- to perimenopause or from peri- to postmenopause.”
Point 9: Line 253, “Previous cross-sectional studies have shown that women of reproductive age have lower iron biomarkers than middle-aged and elderly women [19, 20].” This statement is not clear because not all bio markers were low in women of reproductive age. For example: soluble transferrin receptor (figure 3E).
Response 9: We thank the reviewer for this excellent comment. We revised the statement.
Page 9, line 263- 264 “Previous cross-sectional studies have shown that women of reproductive age have lower ferritin levels than middle-aged and older women [5, 6].”
Point 10: The authors have rightly mentioned in limitations in line 302, “we did not have information on which day of the menstrual cycle iron was measured and we could not take this into account in our analysis.” As iron levels can change drastically in menstruating women, this is an important parameter whether the blood was drawn before or after mensuration and the day of blood collection should have been ideally kept constant throughout study.
Response 10: We thank the reviewer of this great feedback and confirmation.
Point 11: Additionally, I would like to understand why authors didn’t use the new age technology like ICPMS or AAS for serum iron measurement.
Response 11: We thank the reviewer for the comment, but colorimetic method of serum iron is standard practice in our University Hospital. It is an excellent method to determine serum iron, and definitely at lower costs than ICPMS or AAS for serum iron, especially since serum iron is measured on a highly frequent basis in clinical practice. For our articles, we try to adhere as much as possible to the same measurement method which is used in clinical practice as these values reflect the values than clinicians see in the judgment of patients.
Point 12: Comments on the Quality of English Language
The English needs to be improved to improve the overall quality of manuscript and to make it readable. It is really difficult to interpret why authors want to say at some points.
Response 12: We thank the reviewer for this comment. We edited the text again and asked a native to review it, which can be seen the changes in red light.
- Milman, N., M. Kirchhoff, and T. Jørgensen, Iron status markers, serum ferritin and hemoglobin in 1359 Danish women in relation to menstruation, hormonal contraception, parity, and postmenopausal hormone treatment. Ann Hematol, 1992. 65(2): p. 96-102.
- Zacharski, L.R., et al., Association of age, sex, and race with body iron stores in adults: analysis of NHANES III data. Am Heart J, 2000. 140(1): p. 98-104.
- Milman, N. and M. Kirchhoff, Iron stores in 1359, 30- to 60-year-old Danish women: evaluation by serum ferritin and hemoglobin. Ann Hematol, 1992. 64(1): p. 22-7.
- Grisotto, G., et al., Menopausal Transition Is Not Associated with Dietary Change in Swiss Women. J Nutr, 2021. 151(5): p. 1269-1276.
- Milman, N., et al., Iron status in Danish women, 1984-1994: a cohort comparison of changes in iron stores and the prevalence of iron deficiency and iron overload. Eur J Haematol, 2003. 71(1): p. 51-61.
- Cho, G.J., et al., Serum ferritin levels are associated with metabolic syndrome in postmenopausal women but not in premenopausal women. Menopause, 2011. 18(10): p. 1120-4.

Round 2
Reviewer 1 Report
The authors have adequately addressed all the points raised.
Author Response
Response: We thank the reviewer for this great feedback.
Reviewer 2 Report
The authors have done a great job and included all the relevant information in updated version. Although, they have explained about the exclusion criteria for 480 subjects but missed to add it in the manuscript text, they should add it.
Author Response
Response: We thank the reviewer for this great feedback and the excellent comment. We added the relevant statement to the text and highlighted it.
Page 2, lines 85- 86 “Therefore, we excluded additional 480 cases out of 5,702 individuals (n=5,222).”
